# Determination of the Optimal Size of Photovoltaic Systems by Using Multi-Criteria Decision-Making Methods

**Guido C. Guerrero-Liquet [1],[*],[†]** , **Santiago Oviedo-Casado [2],[*],[†]** , **J. M. Sánchez-Lozano [3]**,
**M. Socorro García-Cascales [1]** , **Javier Prior [2],[4] and Antonio Urbina [1]**

[1] Department of Electronics, Computers Technology and Projects, Technical University of Cartagena, c/Dr. Fleming, 30202 Cartagena, Spain; socorro.garcia@upct.es (M.S.G.-C.); antonio.urbina@upct.es (A.U.)

[2] Departamento de Física Aplicada, Universidad Politécnica de Cartagena, 30202 Cartagena, Spain; javier.prior@upct.es

[3] Centro Universitario de la Defensa de San Javier, Academia General del Aire, Universidad Politécnica de Cartagena (UPCT), Murcia 30720, Spain; juanmi.sanchez@cud.upct.es

[4] Institute Carlos I for Theoretical and Computational Physics, Universidad de Granada, 18071 Granada, Spain

[*] Correspondence: guidoc.guerrero@upct.es (G.C.G.-L.); santiago.oviedo@upct.es (S.O.-C.); Tel.: +34-968-326-514 (G.C.G.-L.)

[†] These authors contributed equally to this work.

**Abstract:** The diverse socio-economic and environmental impacts related to the setup of a new photovoltaic installation must be weighed carefully in order to reach the best possible solution. Among the different photovoltaic systems, there are several classification criteria, depending on the technology, application, and size of the modules that define them. The size (installed nominal capacity) stands out as an impartial and critical measure in the decision-making process. In this article, we use a multi-criteria decision-making method to analyze the responses of five experts to a detailed questionnaire in which several different criteria are correlated with various photovoltaic installation sizes. The limitation associated with a low number of experts is addressed with a robustness and sensitivity analysis. With this study, we seek first to apply and demonstrate the feasibility of a methodology that combines technical information with multi-criteria decision-making methods. Second, we obtain a clear result that increases the benefits of a forthcoming photovoltaic installation of modules in distributed generation, adding up to one GW total peak power in standard conditions. We observe a consistent result in which smaller photovoltaic modules provide the ideal solution, as this format maximizes the socio-economic benefits of any installation. If a decision has to be taken about the type of photovoltaic plant to be installed, the conclusion is clear: given a certain size, small, easily scalable installations are the best solution for stakeholders, the inhabitants, and the environment.

**Keywords:** solar electricity; photovoltaic systems; distributed generation (DG); multi-criteria decision making (MCDM); analytic network process (ANP); technique for order of preference by similarity to ideal solution (TOPSIS)

---

## 1. Introduction

For many years, industrialized countries have increasingly generated electricity in large centralized facilities. Thus, the electricity generated comes mainly from fossil fuels, nuclear energy, hydroelectric plants, and large solar or wind power plants [1]. However, both climate change awareness and the increasing scarcity and rising prices of fossil energy sources are inducing a shift in the ways

that energy is produced globally. Renewable energy sources are experiencing an important increase in the power installed per year.

The International Council of Large Electrical Systems (CIGRÉ) defines as distributed generation (DG) those generation units with a maximum capacity ranging from a few kW to 100 MW, which are usually connected to the distribution network and are not centrally designed [2]. Also, DG can be defined as the generation of energy by small-scale units that are installed in the distribution systems where the energy is consumed by the end users [3]. The main objective of the distribution network is to provide a reliable and efficient service for consumers while ensuring that voltage levels and quality of supply are within normal parameters [4]. Traditionally, this objective has been achieved through the reinforcement of existing lines and substations, or through the installation of new DG systems [5,6]. Including renewable sources in the DG grid consequently contributes to a cleaner electricity mix [7].

Photovoltaic (PV) solar energy connected to the grid can be expanded as a modular DG. This saves in initial investments, as small systems can be later upgraded to larger ones if needed, thus making their installation by the end user quite practical [8]. In addition, the modular nature of photovoltaic technologies allows a cost per unit of installed power capacity and power conversion efficiency that is almost independent of the size of the installation. Therefore, photovoltaic distributed energy is of great interest for decentralized energy production. The distributed photovoltaic systems have a significant impact on price [9,10]. The total cost of the distributed rooftop solar PV system is in principle more expensive than large-scale solar PV plants, but it has followed a similar price reduction trajectory—especially regarding the cost of the photovoltaic modules—and is nowadays competitive with (or cheaper than) retail electricity prices in many locations (considered as leveled cost of electricity, LCOE) [11]. Contrary to most conventional energy sources, photovoltaic technology has a wide range of applications in a wide range of sizes. For this reason, the size parameters of the DG have to be carefully determined to improve the performance and overall efficiency of the PV. Therefore, the proper size of a distributed power installation is crucial factor for reliability and meeting consumer demand.

Most of the studies aiming to determine the optimal size of photovoltaic installations have focused on small-scale facilities, namely those ranging from solar roofs in the consumer or end user [12]. However, some recent studies emphasize that large solar power plants or large-scale installations are the most recommended, as they guarantee supply needs are covered and reduce greenhouse gas emissions considerably [13]. In our study, the modules composing the optimum installation are required to add up to one GW. We aim to correlate several different economic, technical, and environmental criteria to obtain the ideal module size of a photovoltaic DG installation for each possible application.

It is clear that adequately classifying the size is a crucial aspect when determining the optimal modules of a distributed generation photovoltaic installation [14]. In this article, we have divided these facilities into a small and medium-sized plants category (composed of less than one MW of power units, and between one and five MW respectively), corresponding to those usually installed in homes and buildings [15] (a second category, including larger sizes (>5 MW) but still adding up to one GW is presented in the Supplementary Materials. Both the methodology and the results obtained for this large category are the same as for the main, small, and medium-sized categories).

With regard to the classification of DG units by order of installed power, scientific research currently uses many definitions, including [1]: a few kilowatts up to 50 MW (Electric Power Research Institute), between five and 25 MW (Gas Research Institute), or less than 50–100 MW (CIGRÉ). The categorization by installed power is typically divided between individual systems or power plants [16]. Individual systems are those with a power unit size (or generation capacity) of the order of kW, and power plants are photovoltaic installations that have a power unit around MW and large-scale photovoltaic systems or small-scale photovoltaic systems [17]. Large-scale photovoltaic systems are considered to be solar plants greater than 500 kW, and small-scale photovoltaic systems are considered to be solar installations greater than three kW and less than 500 kW. Here, we will employ the classification proposed by Viral and Khatod in 2012 [18], which is summarized in Table 1.

**Table 1.** Different ratings of distributed generation (Source: [18]). The small and medium sizes constitute the main focus of this article, while the results for the large category are presented in Supplementary Materials A.

| Size | Categories | Power |
|---|---|---|
| Small and medium distributed generation | Microdistributed generation | ~1 W < 5 kW |
| | Small distributed generation | 5 kW < 5 MW |
| Large distributed generation | Medium distributed generation | 5 MW < 50 MW |
| | Large distributed generation | 50 MW < 300 MW |

Distribution engineers need new planning tools to maximize benefits in uncertain scenarios [19]. To analyze the relative influence of what in principle are unrelated, independent, or incomparable criteria, powerful optimization techniques and expert advisory panels are needed [20,21]. Here, we employed a committee composed of five experts with heterogeneous backgrounds (the panel is composed by academics working in renewable energy technology—from Technical University of Cartagena—two experts with experience in photovoltaic system installations working in private companies, and finally an end user of a small PV system), who selected a total of 14 criteria, which were then analyzed employing multi-criteria decision-making methods (MCDMs) [22–24], in search for the best size alternative fulfilling the decision criteria. To overcome the limitation of the low number of experts, which was caused mainly by the length of the questionnaires, we performed a sensitivity analysis implemented in Matlab/Octave software, which asserts the robustness of the method and the optimal scale chosen between the different alternatives. In this article, a MCDM supported by robustness analysis is used for the first time to the problem the size of photovoltaic installations, revealing that, contrary to the common paradigm, installations composed by smaller, easily scalable modules maximize the socio-economic and environmental benefits of photovoltaic energy. The objective of this study is first, to apply and demonstrate the feasibility of a methodology that combines technical information with multiple-criteria decision-making methods, and second, to obtain a clear result aimed at increasing the socioeconomic benefits of photovoltaic installation modules in distributed systems, providing a case example of a study that adds up to one GW of maximum power under standard conditions.

This article is organized as follows. In Section 2, the methodology (i.e., the specific MCDM) employed to analyze the experts' responses and a correlation among the criteria is described in detail. Section 3 applies the MCDM in a case study example that evaluates the optimal size of photovoltaic systems. This section also presents the results obtained, the comparative analysis, and the analysis of sensitivity conclusions. In Section 4, the results and analysis are discussed. Finally, Section 5 presents the conclusions.

## 2. Materials and Methods

A MCDM is a methodology for making complex problem decisions in a systematic and structured way [25], with the objective of providing an effective framework for the classification and selection of one or more options from a set of alternatives [26]. Practical problems are often characterized by several contradictory criteria, and there may be no solution that meets all the criteria simultaneously. The solution has then to be a compromise, according to the preferences of the decision maker [27].

The best-known methods based on multi-criteria decision include the following: analytical hierarchical process (AHP), analytic network process (ANP), elimination and choice expressing reality (ELECTRE), preference ranking organization method for enrichment evaluation (PROMETHEE), technique for order of preference by similarity to ideal solution (TOPSIS), multi-criteria optimization and compromise solution (VIKOR), and decision-making trial and evaluation laboratory (DEMATEL) [28]. MCDMs have been widely applied in the energy fields, such as site selection, or project and equipment evaluation [29–31]. In order to achieve our objective, we combine two widely

used multi-criteria decision methods: the ANP (analytical network process) developed by Saaty [32], and the TOPSIS (technique for preference by similarity to ideal solution) developed by Hwang and Yoon [23]. The evaluation procedure of this study consists of the following phases, which are shown here in Figure 1.

The first step is to identify the multiple criteria that are considered in the decision-making process. Then, the experts determine the relative influence among the different criteria by evaluating the degree of interdependence. After constructing the dependence network, the relative weights of the different criteria are calculated with ANP. Finally, TOPSIS is used to provide a ranking of the alternatives [33].

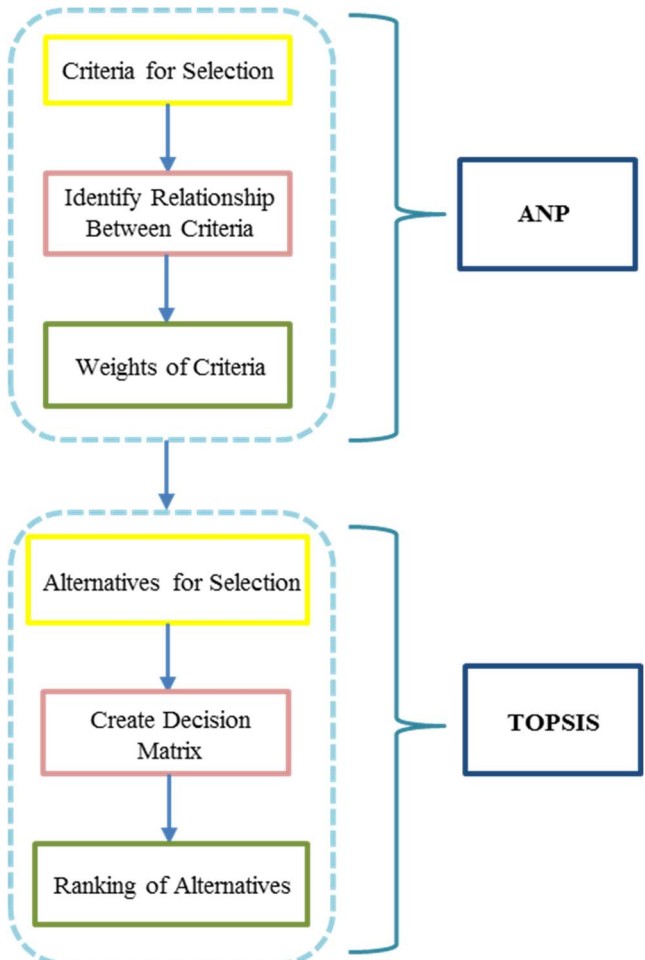

**Figure 1.** Schematic representation of the different phases of a combined analytic network process– technique for order of preference by similarity to ideal solution (ANP-TOPSIS) methodology.

The ANP-TOPSIS combination eliminates many of the ANP procedures as redundant. Calculations and comparisons by additional pairs are avoided in classifying the alternatives [34], hence allowing a solution to be obtained in a shorter time. The ANP-TOPSIS combination has been used in other areas (S.I. B); but in the field of photovoltaic solar energy, it is used for the first time in this study. The reason may be that the ANP methodology has only recently been introduced into the scope of multi-criteria methodology [35]. "Several studies have used the hybrid ANP-TOPSIS to evaluate energy-related problems, for example for power plants [36], [the] evaluation of different biodiesel blends [37] or global approaches to full energy systems at [the] country level [38]."

### 2.1. Analytic Network Process (ANP)

The ANP is a generalization of the analytical hierarchical process (AHP) methodology in which hierarchies are replaced by networks that capture the dependency and feedback within and among elements [22]. Many traditional MCDMs are based on the assumption of independence. However, most situations do not meet the independence condition [26]. Therefore, ANP is divided into two parts. The first consists of a hierarchy of control, or the network of criteria and sub-criteria that controls the interactions. The second is a network of influences between elements and groups [39]. The ANP combines all the possible results in the estimation of the relative influence of the different criteria from which the general priorities derive [29]. Here, the ANP is used to obtain the relative weight of each of the criteria for the proposed model. The advantages that ANP has are as follows. It does not restrict itself to ordering the elements in a hierarchy, it allows collecting relations of interdependence and feedback, and it can handle different natures of criteria or elements. The disadvantages are: a broad comparison scale, a lot of time to determine the system variables, the range inversion problem, and the priority derivation method.

### 2.2. Technique for Order of Preference by Similarity to Ideal Solution (TOPSIS)

The TOPSIS method is a decision model for the classification of preferences by similarity to the ideal solution. The basic principle of the method is the search for an alternative that minimizes the distance with the positive ideal solution [40]. In TOPSIS, the weight of each of the criteria is known a priori (hence ANP). However, in many real situations, clear data are inadequate to model real-life situations, since human judgment is vague, and cannot be estimated with exact numerical values. In a fuzzy environment such as the one used throughout this article, Fuzzy TOPSIS could be used to take into account the diffuse variables. However, as it is combined with ANP to obtain the influence of the experts, it does not directly apply here [26]. With ANP-TOPSIS, the preferences of the decision makers show clear values, and Saaty [32] recommended not using Fuzzy ANP, because it does not provide a viable solution. This method allows searching for the better alternatives for each criterion exposed in a simple mathematical form, with the relative importance of the weights incorporated in the comparison procedures [26]. Here, TOPSIS is used to obtain the values of the alternatives. The advantages of the TOPSIS method are as follows. It is relatively simple and fast; it is able to deal with the problem of reversion and identify the best alternative quickly; pair wise comparisons are avoided; its logic is rational; and importance weights are incorporated in the comparison procedures. The disadvantages are as follows. The investment range is insensitive to the number of alternatives; it suffers from the inherent problem of assigning reliable subjective preferences with the criterion, and exhibits poor performance when there are a very limited number of attributes.

### 2.3. ANP-TOPSIS Methodology

In this study, a hybrid multi-criteria decision method is selected; specifically, ANP and TOPSIS are combined, because these methods are able to handle multiple criterion problems with innumerable experts obtaining the influence of the criteria and the evaluation of the alternatives in a considerably short time, taking into account the binary relations and the type of preference as indicated by different studies such as Wątróbski et al. [41]. Thus, the ANP-TOPSIS methodology proposed is composed of the 10 steps presented in Figure 2.

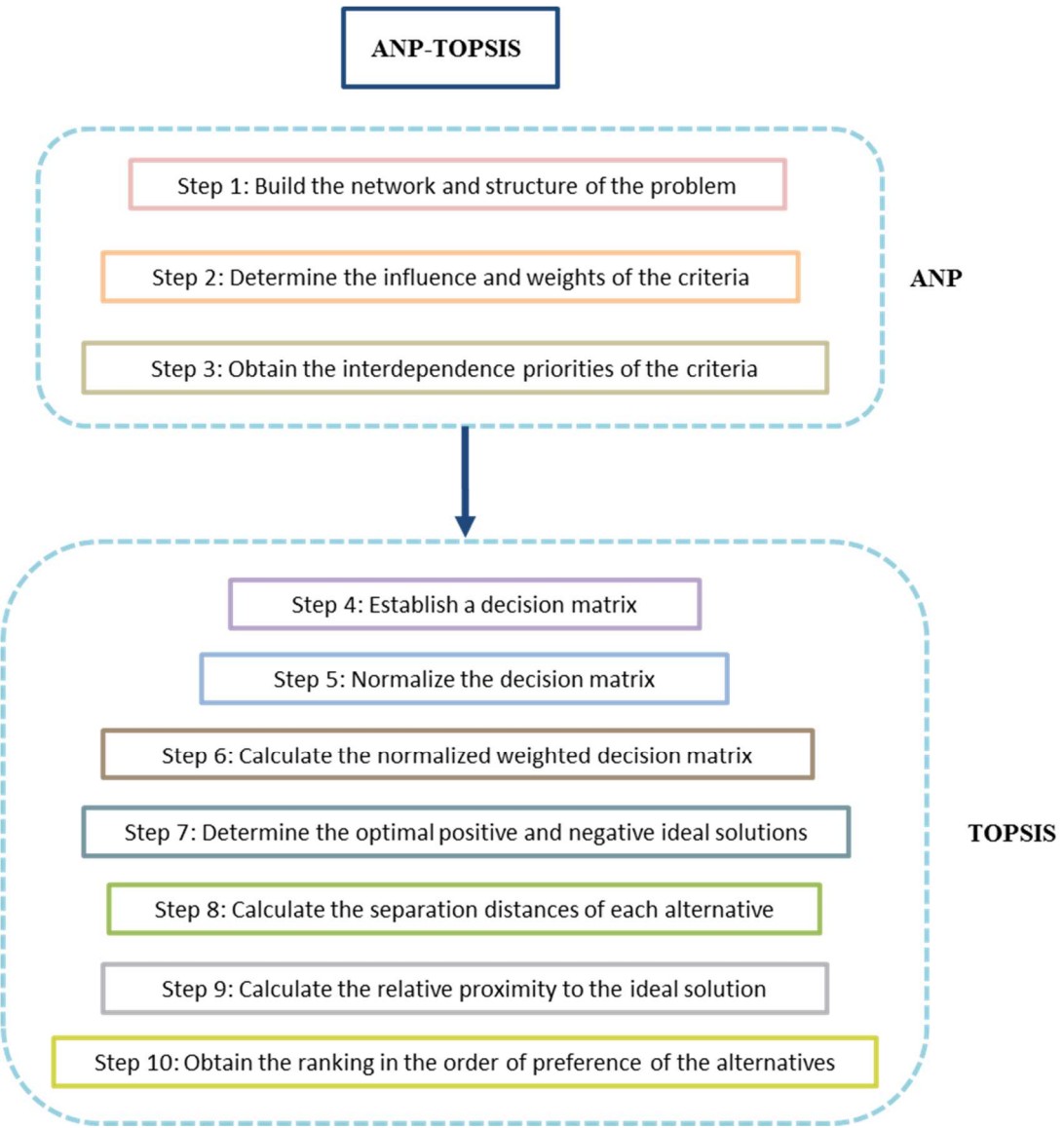

**Figure 2.** The 10 different steps of ANP-TOPSIS as implemented through this article.

● Step 1—Build the network and structure of the problem

The network will be determined by the opinion of the decision panel through the exchange of ideas or other appropriate methods, such as reviews of the literature. The elements of the network (criteria, sub-criteria, and alternatives) are identified and grouped. Then, the network of influences is created through the matrix of interaction domination:

$$
\begin{array}{c}
\\
C_j \\
\begin{array}{ccccc}
& e_{j,1} & e_{j,2} & \cdots & e_{j,n} \\
C_j \quad e_{i,1} & a_{i_1,j_1} & a_{i_1,j_2} & \cdots & a_{i_1,j_n} \\
e_{i,2} & a_{i_2,j_1} & a_{i_2,j_2} & \cdots & a_{i_2,j_n} \\
\cdots & \cdots & \cdots & \cdots & \cdots \\
e_{i,m} & a_{i_m,j_1} & a_{i_m,j_2} & \cdots & a_{i_m,j_n}
\end{array}
\end{array}
\tag{1}
$$

where $a_{ii,jj}$ represents the influence of the element $e_{ii}$ on the element $e_{jj}$, $a_{ii,jj}$ = one, which represents the influence of the element $e_{ii}$ on the $e_{jj}$, and $a_{ii,jj}$ = zero where $e_{ii}$ does not influence on the $e_{jj}$.

● Step 2—Determine the influence and weights of the criteria

The priorities of the criteria are calculated from the paired comparison matrix according to the identified interdependence, with a scale from one to nine [32]. One represents the indifference between the two criteria, and nine means that the criterion considered is extremely important or influential to the compared criterion. First, the influence priorities are calculated:

$$
\begin{matrix}
e_{1,1} & e_{2,1} & e_{2,3} & e_{2,4} \\
e_{2,1} & 1 & r_{1,2} & r_{1,3} \\
e_{2,3} & 1/r_{2,1} & 1 & r_{2,3} \\
e_{2,4} & 1/r_{1,3} & 1/r_{2,3} & 1
\end{matrix}
\tag{2}
$$

Second, the importance priorities of the criteria and sub-criteria are calculated:

$$
w_n = \begin{pmatrix}
1 & r_{12} & \cdots & r_{1n} \\
r_{21} & 1 & \cdots & r_{2n} \\
\cdots & \cdots & \cdots & \cdots \\
r_{n1} & r_{n2} & \cdots & 1
\end{pmatrix} \quad r_{ij} \cdot r_{ji} = 1
\tag{3}
$$

It is possible to measure the consistency of the decision makers' judgments. The ANP provides a measure through the consistency ratio (CR), which is an indicator of the model's reliability [39].

● Step 3—Obtain the interdependence priorities of the criteria

The relative importance of the criteria that consider interdependence is obtained by synthesizing the results of the priorities as follows [42]:

$$
w_c = \begin{pmatrix} C_1 \\ C_2 \\ \cdots \\ C_n \end{pmatrix} = \begin{pmatrix}
1 & e_{12} & \cdots & e_{1n} \\
e_{21} & 1 & \cdots & e_{2n} \\
\cdots & \cdots & \cdots & \cdots \\
e_{n1} & e_{n2} & \cdots & 1
\end{pmatrix} \cdot \begin{pmatrix} w_1 \\ w_2 \\ \cdots \\ w_n \end{pmatrix}
\tag{4}
$$

● Step 4—Establish a decision matrix

Establish a decision matrix for alternative performance. The structure of the matrix can be expressed as follows:

$$
\begin{matrix}
 & w_{c_1} & w_{c_2} & \cdots & w_{c_j} & \cdots & w_{c_n} \\
 & c_1 & c_2 & \cdots & c_j & \cdots & c_n \\
a_1 & x_{1,1} & x_{1,2} & \cdots & x_{1,j} & \cdots & x_{1,n} \\
a_2 & x_{2,1} & x_{2,2} & \cdots & x_{2,j} & \cdots & x_{2,n} \\
\cdots & \cdots & \cdots & \cdots & \cdots & \cdots & \cdots \\
a_n & x_{m,1} & x_{m,2} & \cdots & x_{m,j} & \cdots & x_{m,n}
\end{matrix}
\tag{5}
$$

where $x_{ij}$ represents the value of the alternative $A_i$ with respect to the criterion $C_j$ y, and $W = [w_{c1}, w_{c2}, ..., w_{cn}]$ is the vector of weights associated with the criteria.

● Step 5—Normalize the decision matrix

The associated normalized decision matrix is obtained. For this, the values of each criterion are divided by the norm so that the scale is the same for all the criteria.

$$
\overline{n}_{ij} = \frac{x_{ij}}{\sqrt{\sum_{i=1}^{m}(x_{ij})^2}} \quad j = 1, 2, \ldots, n; \ i = 1, 2, \ldots, m
\tag{6}
$$

● Step 6—Calculate the normalized weighted decision matrix

Each weighted normalized value of the matrix is calculated as the product between each $w_j$ for each $ij$, as expressed in Equation (7).

$$\overline{v}_{ij} = w_{ij}x_{ij} \; j = 1, 2, \ldots, n; \; i = 1, 2, \ldots, m \tag{7}$$

● Step 7—Determine the optimal positive and negative ideal solutions (PIS and NIS, respectively)

The set of positive ($p$) ideal values and the adjusted negative ($m$) ideal value are determined as follows:

$$\overline{A}^p = \left\{\overline{v}_1^p \ldots \overline{v}_n^p\right\} = \left\{\left(\max_i \overline{v}_{ij}, j \in J\right)\left(\min_i \overline{v}_{ij}, j \in J\prime\right)\right\} \text{for j, i} = 1, 2, \ldots, m \tag{8}$$

$$\overline{A}^m = \left\{\overline{v}_1^m \ldots \overline{v}_n^m\right\} = \left\{\left(\min_i \overline{v}_{ij}, j \in J\right)\left(\max_i \overline{v}_{ij}, j \in J\prime\right)\right\} \text{for j, i} = 1, 2, \ldots, m \tag{9}$$

where $J$ is associated with the criteria that indicate the profits or benefits, and $J'$ is associated with the criteria indicating costs or losses.

● Step 8—Calculate the separation distances of each alternative

Calculation of the separation of each alternative with respect to the PIS and NIS, respectively:

$$\overline{d}_i^p = \left\{\sum_{j=1}^n \left(\overline{v}_{ij} - \overline{v}_j^p\right)\right\}^{1/2} \text{for i} = 1, 2, \ldots, m \tag{10}$$

$$\overline{d}_i^m = \left\{\sum_{j=1}^n \left(\overline{v}_{ij} - \overline{v}_j^m\right)\right\}^{1/2} \text{for i} = 1, 2, \ldots, m \tag{11}$$

● Step 9—Calculate the relative proximity to the ideal solution

The calculation of the relative proximity of each alternative to the PIS and NIS using the proximity index:

$$\overline{R}_i = \frac{\overline{d}_i^m}{\overline{d}_i^p} + \overline{d}_i^m \text{ for i} = 1, 2, \ldots, m \tag{12}$$

where the $R_i$ = one value is between zero and one. The closer that the $R_i = 1$ value is to one, the higher the priority of the $i$-th alternative.

● Step 10—Obtain the ranking in the order of preference of the alternatives

Classify the best alternatives according to $R_i$ in descending order.

## 3. Results and Analysis of the Case Study

### 3.1. Application of the Proposed Methodology to the Evaluation of the Optimal Size of Photovoltaic Systems

Below, we provide an example of a case study where the procedure applied to the proposed methodology in a photovoltaic system for distributed generation no greater than one GW of energy is shown. We use ANP-TOPSIS to tackle the question of the size of the photovoltaic installations, according to the classification shown in the previous section (Table 1): small/medium and large-scale (shown in the S.I. A) photovoltaic systems for distributed generation. For this analysis, a team composed of five experts belonging to the photovoltaic energy sector and representing areas such as installation engineers, academics, researchers, and users were interviewed using a detailed questionnaire. Three question sessions were held for each expert per case. In the first session, the criteria and sub-criteria were selected, and the relative influence of each of the sub-criteria on

one another was answered by the experts, leading to the matrix of interaction domination (step 1 of ANP-TOPSIS in Figure 2), composed of ones and zeroes, depending on whether a sub-criteria was influenced by another, or not.

The second session questioned both the importance and influence of the criteria among themselves, asking whether a criterion is inconsequential (one) or extremely important (nine) to another (step 2 of ANP-TOPSIS in Figure 2). The geometric mean of each of the resultant matrices leads to the importance priorities among the criteria and sub-criteria (step 3 of ANP-TOPSIS in Figure 2), which when multiplied by the interaction domination matrix, gives us the relative weights of each of the sub-criteria. This will then be used to evaluate the alternatives. In this step, consistency measures are calculated to assert the uniformity of each expert.

In the last session, the experts valued the qualitative criteria for each alternative (step 4 of ANP-TOPSIS in Figure 2), which was then normalized (step 5) and weighted (step 6), i.e., each alternative was composed by 14 values corresponding to each of the sub-criterion, and was multiplied by the corresponding weight calculated in the previous steps.

Finally, to decide the ranking among the alternatives, the ideal positive and negative values (step 7) were calculated from the best possible positive/negatively-weighted alternatives; then, the geometric distance of each of the alternatives to the ideals were calculated (step 8), and with it the relative proximity to the ideal solution (step 9). Finally, a ranking of alternatives according to the relative proximity to the ideal was established (step 10). The following subsections present the system components (criteria, factors, and alternatives), results, and the comparative analysis.

This section may be divided by subheadings. It should provide a concise and precise description of the experimental results, including their interpretation as well as the experimental conclusions that can be drawn.

### 3.2. Criteria and Factors to Consider for the Optimal Size in PV

First, four large main groups of criteria have been established. They are the general criteria that determine the optimum size of the photovoltaic system: technical, economic, environmental, and social criteria. Fourteen sub-criteria are selected from almost 70 determining factors that affect the optimal size of a distributed generation photovoltaic system (Table 2) that recent studies have shown [43–47], according to the criterion to which they belonged. These sub-criteria were divided into quantitative and qualitative according to the characteristics of measurement in each case.

**Table 2.** Parameters determining the small or medium size of a distributed generation photovoltaic system.

| Criteria | Sub-Criteria | Measurement Characteristics | Factors That Make It Up |
|---|---|---|---|
| Technical (CT) | Connection to the Network (CNK) | Qualitative | Overvoltage, Additional transmission lines, Transmission and distribution losses, Intermittency in generation, Connection facilities, Distance to transformation substations |
| | Geolocation (GEO) | Qualitative | Geographic location, Solar resource available, Site surface, Performance ratio, Angle of inclination, Orientation |
| | Annual Power Loss (APL) | Qualitative | Loss of annual power output due to degradation of the modules |
| | Functionality of the System (FUS) | Qualitative | Technical and operational constraints, System reliability, System degradation |

**Table 2.** *Cont.*

| Criteria | Sub-Criteria | Measurement Characteristics | Factors That Make It Up |
|---|---|---|---|
| Economic (CE) | Economic Costs (EC) | Quantitative | Operating costs, Investment cost, Maintenance cost, Cost of electricity supplied by the network, Saving transmission |
| | Economic Barriers (EB) | Qualitative | Barriers to financing, Barriers in hiring |
| | Incentives and Economic Profitability (IEP) | Qualitative | Possible aid and tax relief, Additional income due to possible emission reduction, Internal rate of return (IRR), Net Present Value (NPV), |
| Environmental (CM) | Physical Impacts on the Ground (PIG) | Qualitative | Natural restrictions in the topography of the land, Land availability, Environment |
| | Environmental viability (EV) | Qualitative | Environmental impact, Reducing greenhouse gas emissions, Recovery end of life photovoltaic project, Legal restrictions on environmental protection |
| Social (CS) | Generated Employment (GE) | Quantitative | Employment generated in the construction phase, Employment generated in the operation phase, Employment generated in the dismantling and recycling phase |
| | Regulatory framework (RE) | Qualitative | Regulatory framework |
| | Socio-political Perception of the Population (SPP) | Qualitative | Acceptance of the community and perceived equity, Public perception of information, Socio-political impacts, Poverty alleviation and reduction of inequalities |
| | Socio-economic Viability (SV) | Qualitative | Promotion of energy savings and awareness of environmental problems, Development of local infrastructure, Production of goods and services, Economic development |
| | Electric Supply Service (ESS) | Qualitative | Energy utilization, Local dispatch ability, Degree of user satisfaction, Demand coverage |

With the knowledge of the criteria and the factors that compose them, the panel of specialists identified their mutual influence to build the network and the structure of the problem (Figure 3).

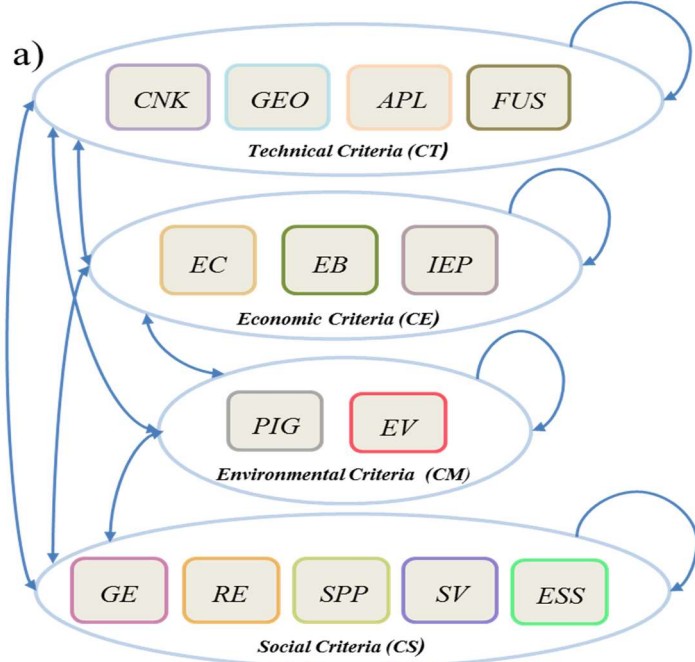

**Figure 3.** The interrelation among criteria used for the evaluation of small and medium-scale systems. Notice that the arrows indicate that each cluster has a relationship with criteria within other clusters and within the same cluster. The same relation has been used for large systems (see the Supplementary Information).

### 3.3. Results of the Model in Small and Medium-Sized Photovoltaic Systems

In order to obtain the weight of the sub-criteria ($w_c$) of the small and medium size of the PV systems with DG, we carried out two questionnaire stages. Steps 2 and 3 of the ANP-TOPSIS are applied. In this stage, the consistency of each of the experts is analyzed. The result of the weights helps us evaluate the alternatives in the following steps (these results are presented in detail on S.I. B).

To determine the alternatives, we have a size range from five kWp to five MWp, according to the small and medium size that we have determined of the photovoltaic systems. The number of systems in each alternative should add up to one GWp of installed capacity; therefore, the different size alternatives are comprised by the different number of independent systems: the smaller the size of the system, the larger number that need to be installed in that category. The following four groups have been weighted as alternatives:

- Alternative 1 (AS1)—200,000 photovoltaic systems with five kWp of power. These facilities would be located in multi-family housing for self-consumption.
- Alternative 2 (AS2)—2000 photovoltaic systems with 500 kWp of power. These facilities would be located in the roofs and parking areas of medium-sized companies.
- Alternative 3 (AS3)—1000 photovoltaic systems of one MWp of power. Installations located on decks and car parks of large companies with an area larger than 5000 square meters
- Alternative 4 (AS4)—200 photovoltaic systems with five MWp of power. Installations located in industrial areas that contain the necessary surface according to established regulations.

The alternatives can be considered independent systems (all the same size within the alternative) or as groups of systems that add up to the maximum size considered in this alternative, therefore providing a partial mixed solution within each alternative. For example, in alternative 1, a few systems of five kWp can be grouped and considered as a larger system (limited by the maximum of 500 kWp, which is the minimum for the following alternative). The results of the decision matrix step 4 of Section 2.2 can be seen in S.I. B. The data of the quantitative sub-criteria economic costs (EC)

and generated employment (GE) were weighted based on a study published by the International Renewable Energy Agency [48]. The other sub-criteria are qualitative, and their value was established from the answers given by the experts in the last stage of questionnaires. Applying the remaining steps of ANP-TOPSIS, we obtain the distance of each of the alternatives to the ideal solution, which permits their classification in preference order, as shown in Figure 4 and Table 3.

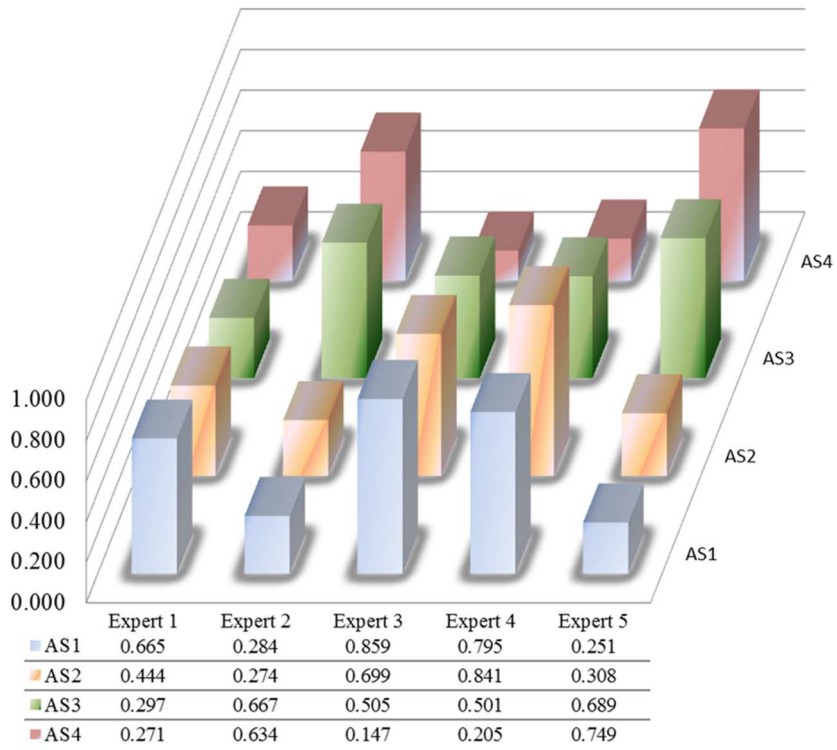

|  | Expert 1 | Expert 2 | Expert 3 | Expert 4 | Expert 5 |
| --- | --- | --- | --- | --- | --- |
| AS1 | 0.665 | 0.284 | 0.859 | 0.795 | 0.251 |
| AS2 | 0.444 | 0.274 | 0.699 | 0.841 | 0.308 |
| AS3 | 0.297 | 0.667 | 0.505 | 0.501 | 0.689 |
| AS4 | 0.271 | 0.634 | 0.147 | 0.205 | 0.749 |

**Figure 4.** Relative proximity of each of the alternatives composing the size choices in small and medium distributed photovoltaic systems.

**Table 3.** Ranking of the small and medium-scale alternative cases as classified by the relative distance calculated from the analysis of each expert.

| Experts | Expert 1 | Expert 2 | Expert 3 | Expert 4 | Expert 5 |
| --- | --- | --- | --- | --- | --- |
| Ranking | AS1 > AS2 > AS3 > AS4 | AS3 > AS4 > AS1 > AS2 | AS1 > AS2 > AS3 > A4 | AS2 > AS1 > AS3 > AS4 | AS4 > AS3 > AS2 > AS1 |

### 3.4. Comparative Analysis

The comparison of the results obtained from the alternatives is shown in Table 3, where the ranking of sizes for each of the experts and the trend that finally prevails is presented.

The most repeated trend is A1 > A2 > A3 > A4 (the same result is found in the large installations case, see Supplementary Information), while no other trend is repeated. The deviations produced from this trend can be analyzed in terms of sensitivity to human errors, as we will explore in the next section. The predominant alternative is therefore 200,000 small installations of five kWp for the small and medium-sized groups, selecting thus the smaller size within the available range. These results suggest that for a given output power, smaller size modules optimize the distributed generation in photovoltaic systems. The same methodology has been applied to evaluate the large photovoltaic systems, and the result is again that the systems with a smaller size in this category are the optimum solution. Details of the parameters and results are presented in the Supplementary Information for the evaluation of large systems. To confirm this trend and the robustness of the results, we carried out a sensitivity analysis that is shown and discussed in what follows.

*3.5. Sensitivity Analysis of the Results*

The results obtained from the ANP-TOPSIS direct analysis show that these results are pretty much consistent for all experts. In this section, we will analyze the robustness of these results against changes and errors both in the evaluation of alternatives and the relative weights assigned to each of the sub-criteria by each of the experts. To do so, we introduce percentage variations on the values provided by the experts in each of the 14 sub-criteria composing the four different alternatives proposed, for both the small–medium and the large solar plant cases (the latter shown in the S.I.). Thus, we simulate errors that the experts might have committed in assessing the value assigned to each of the sub-criteria in each of the alternatives. In that way, we confirm the strength of the decision and the method used to reach it, and also identify the weak points and the most sensitive sub-criteria, whose values might have to be double-checked.

Given the structure of the ANP-TOPSIS decision method, percentage variation analysis is the same for both the final relative weights assigned to each of the sub-criteria and the values provided for each alternative. In the ANP-TOPSIS method, the normalized final weights obtained after correlating the criteria and sub-criteria are each multiplied by the corresponding normalized value of each sub-criteria in each alternative. This particularly means that any variation produced in either of the weights (alternatives) will spread and diffuse to all the weights when the normalization procedure is applied. Moreover, correlating the alternatives with the weights means that it does not matter where the error is produced, and the results shown in Figure 5 are valid and equivalent for both variations introduced in the weights and the alternatives.

Figure 5a shows the percentage variation of either weight or alternative at which the original ideal solution changes. Results are shown for variations in each of the sub-criteria and for each of the experts. We have extended the variations up to 1000% error in order to show on the one hand, real changes, and on the other, the extreme robustness of the ANP-TOPSIS method, as it is very unlikely that an error of 1000% is produced. In particular, the smallest error for which a change in the ideal alternative appeared is 33%, which was observed for expert 1. Furthermore, only three of the variations produced a change within the 100% error, displaying the robustness of the method. Moreover, such changes appear in experts for which the solution differed from the common A1 > A2 > A3 > A4 solution, and the tendency is for the changes induced by errors to drift toward the A1 > A2 > A3 > A4 solution. With this analysis, we gain insight on the influence that each of the weights has on the final solution. However, it is useful to study the possibility of joint errors as well.

In Figure 5b, we have again performed a sensitivity analysis by varying each of the weights and adding an error, in this case up to 100%. The difference is that here, all of the other weights (or alternatives) are allowed to have an error equal or smaller than the one that is being analyzed. To do so, 1000 realizations of random errors for each of the percentages are calculated and averaged to obtain how the ideal solution changes with error. The percentage at which the ideal solution changes is represented in the heat map on Figure 5b. We see that for most of the sub-criteria, an error higher than 100% in all of the sub-criteria is necessary in order to see a change in the ideal solution. Moreover, in any case, such an error has to be higher than 15%. This again demonstrates the robustness of the method that we have employed and the fiability of the results obtained. This analysis shows as well the degree of interrelation existing among sub-criteria, demonstrating that except for sub-criteria three, where the sensitivity to errors in the other sub-criteria produces a change in the ideal solution for all of the experts, the choice of interrelations by the experts that leads to the relative weights is well justified, as the interrelations are already taken into account, and they do not exert more influence. In that sense, the sensitivity analysis demonstrates that the results obtained from the experts' analysis is robust.

The robustness of the ANP-TOPSIS method comes mainly from any error that is produced regarding sub-criteria being normalized and thus spreading on all the sub-criteria, causing a shift without order change in the results when the whole ANP-TOPSIS is implemented. In fact, if an error is introduced in more than one of the alternatives (weights), the results show that the percentage variation needed to see a change in the ideal solution increases for most of the sub-criteria. Correlations between

sub-criteria seem to be small, and are only important whenever it coincides that both weights are similar, and both correspond to either a maximum or minimum in the distance to the ideal solution. In general, this means that the possible correlations tend just to reinforce the final solution instead of changing it, which is again due to the normalization procedure introduced.

In conclusion, the robustness analysis shows that solutions are resilient to errors, both in the valuation of alternatives and in the comparison of the relative importance of the different sub-criteria, and demonstrates the capabilities and power of the ANP-TOPSIS method.

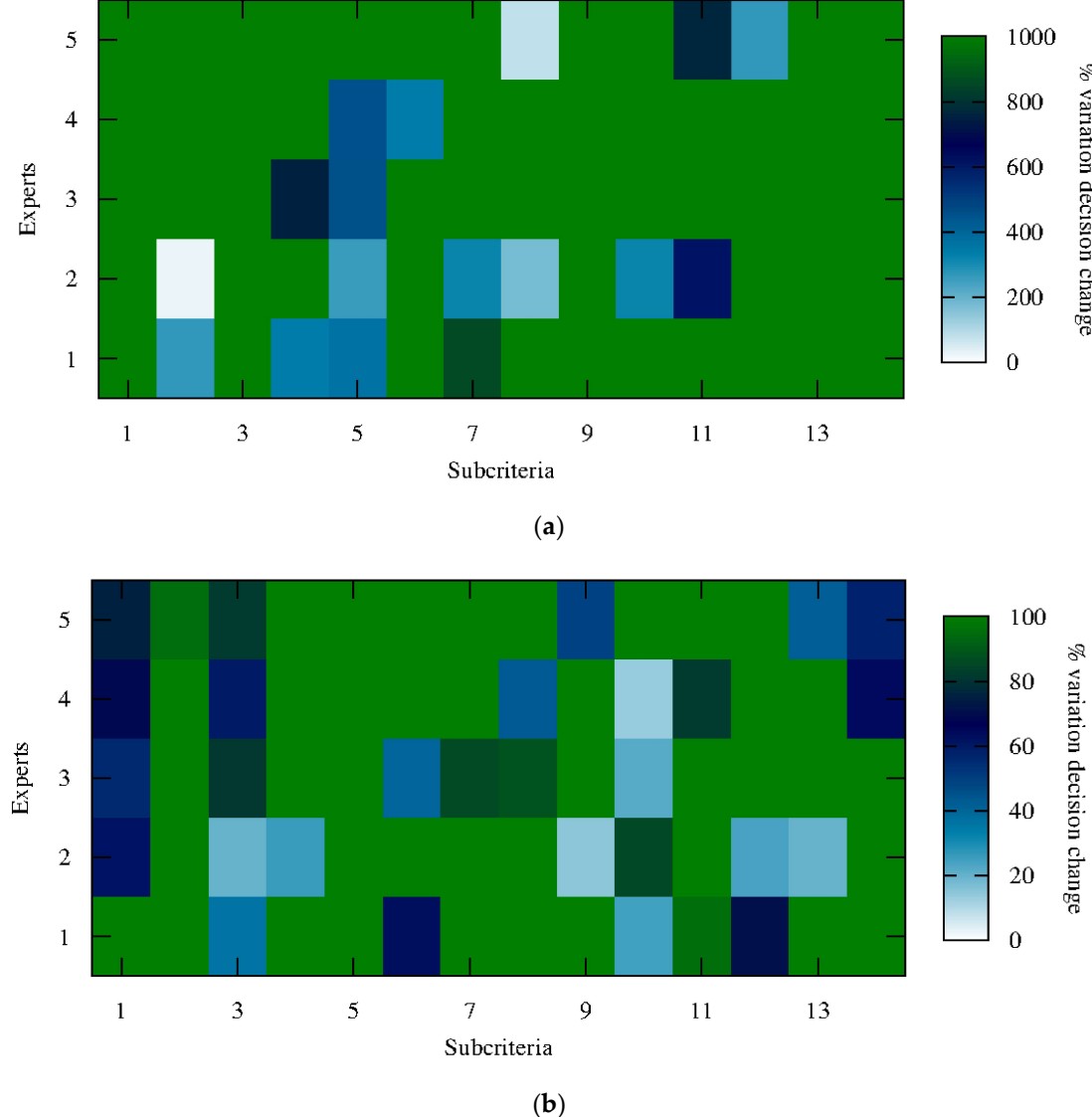

**Figure 5.** (**a**) Percentage variation of the individual sub-criteria weights at which the ideal solution changes for the case of small and medium-scale distributed photovoltaic systems. (**b**) Average of 1000 realizations of percentage variation of the individual sub-criteria together with random errors in all the other sub-criteria of a size that is equal or less than the percentage variation in the considered sub-criteria. Each point shows the percentage at which each interviewee's ideal solution changes for the case of small and medium-scale distributed photovoltaic systems.

## 4. Discussion

Distributed generation photovoltaic plants can be defined through the installed capacity. Provided a desired capacity, the PV plant can be constructed out of modules whose size range from small, domestic installations, to huge solar parks. Moreover, any installed single system can always be

scaled up, thus benefiting from previous investments in the same location due to the flexibility of the technology. Therefore, the size of the system is, unlike other energy technologies, a free parameter that stakeholders (including end users, investors, and policy makers) have to decide by themselves, with few technical limitations and scant constraints. In this article, we have presented a possible classification of the different PV module sizes according to power capacity and target installation location, ranging from a few kW to MW and private, mostly house installations to industrial-size, large solar plants.

Determining the optimal module size for a given power capacity is a highly non-trivial problem, where several technical, socio-economic, and environmental parameters can be assigned to each size. In this article, a methodology to obtain the optimum decision regarding the size of the system in terms of the best socio-economic impacts is proposed and tested. For that purpose, panels of experts have been interviewed in order to assign the qualitative socio-economic and environmental parameters to different PV systems according to its size, such as for example, the amount of new jobs created or the physical impact on the ground. Once the deciding criteria are selected, and the weights are correspondingly assigned, we have, in what to our knowledge is a first in photovoltaic technologies, applied a combination of ANP and TOPSIS multi-criteria decision-making methods to evaluate the different PV systems and obtain the best-fitting solution within each size group. This is achieved by combining the relative weights of each of the different criteria and sub-criteria, with the either numerical values for each sub-criterion (e.g., jobs generated) or the relative importance of the sub-criteria. In addition, we have implemented a sensitivity and robustness analysis of the method by performing a cross-correlation and error study, which indicated that the proposed solution is robust, there is no superfluous correlation between parameters, and that the methodology itself is built so that possible errors tend to smooth out and become relatively irrelevant.

## 5. Conclusions

The results obtained show clearly that within each size category (i.e., small and medium versus large systems), the solution that optimizes both the socio-economic and the environmental impacts is the smallest one. This is an important conclusion: the best use of investment regarding the socio-economic impacts of the future PV facilities is to design and build smaller and more distributed PV systems instead of very large plants, which seems to be the actual tendency. Moreover, the decision-making support methodology for this study used a combination of ANP and TOPSIS, which has provided a clear and robust result. This methodology can be extended to other PV studies (examining the advantages of different technologies instead of comparing the different sizes, for example).

In conclusion, initial investment in small systems is recommended. Furthermore, future investment could also be used to expand those already existing installed systems in an optimal way by using the previous photovoltaic plant. Thus, this work is a conscientious effort toward demonstrating—with robust tools and strong arguments—that the policy of "the larger the better" is not necessarily the best, neither for stakeholders nor for the environment. The community in general can benefit greatly from using sophisticated decision methods and smaller photovoltaic installations.

In the aforementioned direction, future work should address the question of the number of experts. It is clear that given the size of the sample and the technicality of the topic, finding suitable experts is a hard task. The sensitivity analysis provides a clue to resolving the issue. It is clear that both the correlation and the relative influence among sub-criteria varies greatly, having almost totally inconsequential sub-criteria. This could be ascertained from the responses of just a handful of experts, thus simplifying the questionnaires for increasing experts samples. Moreover, once a few different experts are known, simulations of experts via learning algorithms can supply the necessary variations that provide robustness to the results by identifying the stronger trends. From the analytical point of view, it would be interesting to explore the TOPSIS group classification in future studies, in order to learn whether the group classification is the same as the most repeated trend (A1 > A2 > A3 > A4).

Another line of future research would be to apply this methodology to specific regions, identifying and solving the particular needs of geographically localized places, which will require knowledge from local experts and specific sub-criteria.

**Supplementary Materials:** The following are available online at http://www.mdpi.com/2071-1050/10/12/4594/s1, SA: Large-scale PV system evaluation; SB: Specific ANP-TOPSIS references; SC: Full numerical results of the evaluation of small–medium and large PV systems.

**Author Contributions:** G.C.G.-L. and S.O.-C. have collected, processed and interpreted data, and have written the manuscript. J.M.S.-L., M.S.G.-C. and A.U. assisted in data interpretation, manuscript evaluation and editing. J.P. and A.U. supervised the development of the work and assisted in data interpretation.

**Funding:** This research was done thanks to the financial support from MINECO (SPAIN), including FEDER funds: FIS2015-69512-R and ENE2016-79282-C5-5-R, and from Fundación Séneca (Murcia, Spain) Project No. 19882/GERM/15 and projects TIN2014-55024-P from MINECO (SPAIN) P11-TIC-8001 and TIN2017-86647-P from Junta de Andalucía (including FEDER funds) and project FIS2015-69512-R from MINECO (SPAIN) and a doctoral scholarship from MESCYT (Dominican Republic) with the contract No. BIM-434-2017, respectively.

**Conflicts of Interest:** The authors declare no conflict of interest. The funders had no role in the design of the study; in the collection, analyses, or interpretation of data; in the writing of the manuscript, or in the decision to publish the results.

## Abbreviations

| | |
|---|---|
| ANP | Analytic Network Process |
| CIGRÉ | The International Council of Large Electrical Systems |
| DEMATEL | Decision making trial and evaluation laboratory |
| DG | Distributed Generation |
| ELECTRE | Elimination and Choice Expressing Reality |
| GW | gigawatt |
| kWp | kilowatts peak |
| LCOE | Levelled cost of electricity |
| MCDM | Multi-Criteria Decision Making |
| MWp | megawatts peak |
| PV | Photovoltaic solar energy |
| PROMETHEE | Preference ranking organization method for enrichment evaluation |
| TOPSIS | Technique for Order of Preference by Similarity to Ideal Solution |
| VIKOR | Multi-criteria Optimization and Compromise Solution |
| $w_c$ | weight of the sub-criteria |

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
