# Peer review of "Determination of the Optimal Size of Photovoltaic Systems by Using Multi-Criteria Decision-Making Methods"

_sustainability, doi:10.3390/su10124594_

Round 1

Reviewer 1 Report

1) The aim of the article should be clearly stated in the introduction.

2) The ANP and TOPSIS methods should be described in more detail in subsections 2.1 and 2.2. The well known ANP-TOPSIS hybrid approach can be described in section 2.3.

3) The authors should justify the choice of the ANP and TOPSIS methods. Maybe the following article will be helpful [1]:

[1] Generalised framework for multi-criteria method selection. https://doi.org/10.1016/j.omega.2018.07.004

Additionally, the authors should indicate the advantages and disadvantages of the selected methods. In example, since the ANP stems directly from the AHP, it also inherits theoretical weaknesses of the assumptions of the AHP which, above all, are: the rank reversal problem, the priorities derivation method and the comparison scale [2].

[2] Selected Issues of Rank Reversal Problem in ANP Method. https://doi.org/10.1007/978-3-319-28419-4_14

4) The paper should contain a literature review section presenting other approaches and research deals with MCDM applications in the photovoltanic systems assessment. In particular, the review should contain criteria used in the assessment.

5) The authors wrote: ‘However, in many real situations clear data are inadequate to model real-life situations, since human judgement is vague and cannot be estimated with exact numerical values’. Therefore, they should explain why they did not use the Fuzzy TOPSIS method, which is dedicated to problems deals with uncertain and inaccurate evaluation [3].

[3] Online Comparison System with Certain and Uncertain Criteria Based on Multi-criteria Decision Analysis Method. https://doi.org/10.1007/978-3-319-67077-5_56

6) From an analytical point of view, the rankings of various experts should be aggregated in one group ranking as in the case of group TOPSIS [4] and Fuzzy TOPSIS [5] methods.  It will be interesting if the group’s ranking is the same as the most repeated trend (A1>A2>A3>A4).

[4] An extension of TOPSIS for group decision making. https://doi.org/10.1016/j.mcm.2006.03.023

[5] A fuzzy approach for supplier evaluation and selection in supply chain management. https://doi.org/10.1016/j.ijpe.2005.03.009

Author Response

 @page { margin: 0.79in } p { margin-bottom: 0.1in; direction: ltr; color: #000000; ; text-align: left; orphans: 2; widows: 2 } p.western { ;Calibri", serif; ; so-language: es-ES } p.cjk { ;Calibri"; ; so-language: en-US } p.ctl { ; ; so-language: ar-SA } a:link { color: #0000ff }

Response letter: Reviewer 1

Your comments are very valuable and have been useful to improve the article. Thank you for the suggestions and recommendations. Now the article is better organized and focused, with the conclusions more clearly emphasized. Additionally, a detailed methodological explanation has been added regarding ANP-TOPSIS methods. Your comments are constructive, so we focus in detail to correct each of the suggestions and we hope that this new version will meet your criteria for publication. Below there is a detailed description of the corrections applied to each of your suggestions or recommendations:

The aim of the article should be clearly stated in the introduction

We have added a brief description of the general objectives of this research at the end of the introduction in lines 115-119.

The objective of this study is first, to apply and demonstrate the feasibility of a methodology that combines technical information with multiple criteria decision making methods, and second, to obtain a clear result aimed at increasing the socioeconomic benefits of a photovoltaic installation modules in distributed systems providing a case example of a study that adds up to 1 GW of maximum power under standard conditions.

2) The ANP and TOPSIS methods should be described in more detail in subsections 2.1 and 2.2. The well known ANP-TOPSIS hybrid approach can be described in section 2.3.

As you suggest, we have created new subsections for the methodology with a more detailed explanation of the ANP-TOPSIS method, including some additional references. In particular:

In line 157 we created the subsection 2.1 Analytic Network Process (ANP);
in line 171 we created the subsection 2.2 Technique for Order of Preference by Similarity to Ideal Solution (TOPSIS) and In line 192 we created the subsection 2.3 ANP-TOPSIS methodology. Globally, the ANP-TOPSIS hybrid approach has been described in detail in lines 200-264, which have been added to the original manuscript.

In addition, sections 2.1 and 2.2 are complemented with a new paragraph with comments about the advantages and disadvantages of each one of the methodologies (lines 16
6-170 and 183-189 respectively).

3) The authors should justify the choice of the ANP and TOPSIS methods. Maybe the following article will be helpful [1]:

[1] Generalised framework for multi-criteria method selection. https://doi.org/10.1016/j.omega.2018.07.004

After reading carefully [1] (reference 41 in the manuscript) and as recommended in section 2.3 ANP-TOPSIS, we have added a paragraph (lines 191-195) where the choice of the ANP-TOPSIS methodology is justified. We quote this work because it shows us the opportune way to select a multi-criteria methodology.

In this study, a hybrid multi-criteria decision method is selected, specifically ANP and TOPSIS are combined because these methods are able to handle multiple criterion problems with innumerable experts obtaining the influence of the criteria and the evaluation of the alternatives in a considerably short time taking into account the binary relations and the type of preference as indicated by different studies such as [38].

Additionally, the authors should indicate the advantages and disadvantages of the selected methods. In example, since the ANP stems directly from the AHP, it also inherits theoretical weaknesses of the assumptions of the AHP which, above all, are: the rank reversal problem, the priorities derivation method and the comparison scale [2].

[2] Selected Issues of Rank Reversal Problem in ANP Method. https://doi.org/10.1007/978-3-319-28419-4_14

Following this recommendation we indicate and discuss the advantages and disadvantages of each one of the methodologies according to the section in which they are presented. For ANP they are indicated in lines 166-170. For the TOPSIS methodology, the advantages and disadvantages are indicated in lines 183-189.

4) The paper should contain a literature review section presenting other approaches and research deals with MCDM applications in the photovoltaic systems assessment. In particular, the review should contain criteria used in the assessment.

In the article we cite different investigations that deal with the applications MCDM in the evaluation of the photovoltaic systems as they are the references 29, 30,31, and indirectly, also focusing on selection and/or preference of distributed energy generation using photovoltaics, references 3, 4, 6 and 9 a special mention to storage selection (12), plants (13) and a few about some particular locations. A new paragraph has been included, in which the new references 36, 37 and 38 are commented:

"Several studies have used the hybrid ANP-TOPSIS to evaluate energy related problems, for example for power plants [36], evaluation of different biodiesel blends [37] or global approaches to full energy systems at country level [38]."

5) The authors wrote: ‘However, in many real situations clear data are inadequate to model real-life situations, since human judgement is vague and cannot be estimated with exact numerical values’. Therefore, they should explain why they did not use the Fuzzy TOPSIS method, which is dedicated to problems deals with uncertain and inaccurate evaluation [3].

Taking into account this consideration in lines 178-183, we have added an explanation about the motivation not to use the Fuzzy TOPSIS method in this case. The method is usually applied to problems related to uncertain and inaccurate evaluation and we consider that this is not the case for this article. The paragraph included in the new version of the manuscript (lines 176-181) justifies this choice:

"In a fuzzy environment such as this Fuzzy TOPSIS could be used to take into account the diffuse variables but as it is combined with ANP to obtain the influence of the experts does not apply [25]. With ANP-TOPSIS the preferences of the decision makers show clear values and Saaty [33] recommends not using Fuzzy ANP because it does not provide a viable solution.

6) From an analytical point of view, the rankings of various experts should be aggregated in one group ranking as in the case of group TOPSIS [4] and Fuzzy TOPSIS [5] methods.  It will be interesting if the group’s ranking is the same as the most repeated trend (A1>A2>A3>A4).

We agree that it would be very interesting to see what trend results from the point of view of the group classification as in the case of the TOPSIS [4] and TOPSIS [5] group methods according to the recommended investigations. But it is beyond the scope of the present work and due to the complexity of this type of study and the limited space in the writing we do not contemplate these analytical comparisons. However, taking into account how interesting the result could be by applying the TOPSIS group classification, we propose it for future research with a paragraph included in the conclusion section. In particular, in the conclusions section in lines 502-505 we underline the fact that it would be very interesting to add in a future investigation the classification of the TOPSIS group to know if the classification of the group is the same as the most repeated trend (A1> A2> A3 > A4).

Reviewer 2 Report

There is no conclusion in this article.

There are too many references in the paragraph "Materials and methods". There should be mainly results of investigations.

Author Response

 @page { margin: 0.79in } p { margin-bottom: 0.1in; direction: ltr; color: #000000; ; text-align: left; orphans: 2; widows: 2 } p.western { ;Calibri", serif; ; so-language: es-ES } p.cjk { ;Calibri"; ; so-language: en-US } p.ctl { ; ; so-language: ar-SA } a:link { color: #0000ff }

Response letter: Reviewer 2

Thank you for helping to improve the article; your review have helped us to improve the quality of the study presented in the article. Each of the comments has been analyzed by the authors to provide the best possible answers which we explain in detail in the following list of comments.

1-There is no conclusion in this article.

We have included a new section devoted to the conclusions of the article. Now this section summarizes the main results and emphasizes our conclusions. Additionally, we point to future work in the final part of this section.

2-There are too many references in the paragraph "Materials and methods". There should be mainly results of investigations

In response to your suggestions, we have reduced some references from the "Materials and methods" section which has been fully rewritten. Nevertheless, some subsections have been added to explain in more detail the methodology (as suggested by other referee), this inclusion has extended the section and although the global numbers of references have been reduced, some of them still remain. In the new subsections we also highlight the investigations of different areas of science that apply the ANP-TOPSIS methodology. As for example lines 154-156:

"Several studies have used the hybrid ANP-TOPSIS to evaluate energy related problems, for example for power plants [36], evaluation of different biodiesel blends [37] or global approaches to full energy systems at country level [38]."

Supplementary information documents show the references of each of these investigations see (S.I.B).

Reviewer 3 Report

The overall contribution of the paper to the field might be valuable, however, some revisions should be done to increase the quality of the paper.

In the literature review please take a look to for instance "G. Celli, E. Ghiani, S. Mocci and F. Pilo, "A multiobjective evolutionary algorithm for the sizing and siting of distributed generation," in IEEE Transactions on Power Systems, vol. 20, no. 2, pp. 750-757, May 2005", that was one of the pioneering paper dealing with MO techniques applied to power systems optimization
- please provide a case study example to clarify the procedure to a real case study

- the paper should be proofread for better readability and correction of any typo and spelling mistake.

- rewrite the conclusions summarizing main findings of the paper and their usefulness for the scientific/industrial community.

Author Response

 @page { margin: 0.79in } h1 { margin-top: 0.19in; margin-bottom: 0.19in; direction: ltr; color: #000000; ; text-align: left; orphans: 2; widows: 2; page-break-after: auto } h1.western { ;Times New Roman", serif; so-language: es-DO } h1.cjk { ;Times New Roman", serif; ; so-language: es-DO } h1.ctl { ;Times New Roman", serif; ; so-language: ar-SA } p { margin-bottom: 0.1in; direction: ltr; color: #000000; ; text-align: left; orphans: 2; widows: 2 } p.western { ;Calibri", serif; ; so-language: es-ES } p.cjk { ;Calibri"; ; so-language: en-US } p.ctl { ; ; so-language: ar-SA } a:link { color: #0000ff }

Response letter: Reviewer 3

These significant recommendations to modify our original manuscript will strengthen the understanding of the article and increase its impact on the scientific community. Below we describe in detail the corrections applied to each of the suggestions or recommendations:

In the literature review please take a look to for instance "G. Celli, E. Ghiani, S. Mocci and F. Pilo, "A multiobjective evolutionary algorithm for the sizing and siting of distributed generation," in IEEE Transactions on Power Systems, vol. 20, no. 2, pp. 750-757, May 2005", that was one of the pioneering paper dealing with MO techniques applied to power systems optimization.

We have consulted this interesting study which is strongly related to the multi-objective methodology applied in our article and we are glad to find many agreements with the conclusions presented therein. Therefore, we thank you for this valuable recommendation. In the introduction to our work we have added line 103 with a reference to this work. We highlight the need we share to apply the new planning and decision-making tools to the uncertainty scenarios offered by the globally distributed energy system.

Distribution engineers need new planning tools to maximize benefits in the new uncertain scenario [19] (Celli G et al 2005).

- Please provide a case study example to clarify the procedure to a real case study

Taking into account your considerations we have restructured the article and added several subsections to the methodology section. In those added paragraphs, the ANP-TOPSIS methodology is explained in detail. We have also added a case study section, to illustrate the use of the methodology that we are proposing in our article. This fact is highlighted in the first paragraph of this section on lines 265-269. Following this paragraph, we present the evaluation of the optimal size of photovoltaic systems applied to a real case in which 5 experts participate.

Below we provide an example of a case study where the procedure applied to the proposed methodology in a photovoltaic system for distributed generation no greater than 1 GW of energy is shown.

- the paper should be proofread for better readability and correction of any typo and spelling mistake.

In response to this recommendation, we have accomplished a revision of the grammar and spelling, thus improving the English style of the document. Thank you very much for your suggestions that have been used to correct some typographical errors.

- rewrite the conclusions summarizing main findings of the paper and their usefulness for the scientific/industrial community.

We have included a new section in which the conclusions are now presented. This new section summarize the main findings our article and also point to future work in the final part of this section with a focus on applications that may be of interest to the scientific/industrial community.

Round 2

Reviewer 1 Report

I accept explanations and justifications of the authors.

Reviewer 3 Report

The paper has been improved and can be accepted in this form.